

# Initial implementation of the resuscitation quality improvement program in emergency department of a teaching hospital in China

Hui Jiang*, Liang Zong*, Fan Li, Jian Gao, Huadong Zhu, Di Shi and Jihai Liu

Emergency Department, State Key Laboratory of Complex Severe and Rare Diseases, Peking Union Medical College Hospital, Chinese Academy of Medical Science and Peking Union Medical College, Beijing, China
* These authors contributed equally to this work.

## ABSTRACT

**Background:** Cardiopulmonary resuscitation (CPR) skills may decay over time after conventional instructor-led BLS training. The Resuscitation Quality Improvement® (RQI®) program, unlike a conventional basic life support (BLS) course, is implemented through mastery learning and low-dose, high-frequency training strategies to improve CPR competence. We facilitated the RQI program to compare the performance of novices *vs* those with previous BLS training experience before RQI implementation and to obtain their confidence and attitude of the RQI program.

**Methods:** A single-center observational study was conducted from May 9, 2021 to June 25, 2021 in an emergency department of a tertiary hospital. The performance assessment data of both trainees with a previous training experience in conventional BLS course (BLS group) and the novice ones with no prior experience with any BLS training (Non-BLS group) was collected by RQI cart and other outcome variables were rated by online questionnaire. Outcome measurements included chest compression and ventilation in both adult-sized and infant-sized manikins.

**Results:** A total of 149 participants were enrolled. Among them, 103 participants were in BLS group and 46 participants in Non-BLS group. Post RQI training, all the trainees achieved a passing score of 75 or more, and obtained an improvement in CPR performance. The number of attempts to pass RQI for compression and ventilation practice was lower in the BLS group in both adult and infant training sessions ($P < 0.05$). Although the BLS group had a poor baseline, it had fewer trials and the same learning outcomes, and the BLS group had better self-confidence. Trainees were well adapted to the innovative training modality, and satisfaction among all of the participants was high. Only the respondents for non-instructor led training, the satisfaction was low in both groups (72.8% in BLS group *vs* 65.2% in No-BLS group, strongly agreed).

**Conclusion:** Among novices, RQI can provide excellent CPR core skills performance. But for those who had previous BLS training experience, it was able to enhance the efficiency of the skills training with less time consumption. Most trainees obtained good confidence and satisfaction with RQI program, which might be an option for the broad prevalence of BLS training in China.

Corresponding authors
Di Shi, shidi@pumch.cn
Jihai Liu, liujihai1977@aliyun.com

## INTRODUCTION

Globally, there is an urgent increasing need to improve the survival of resuscitation among cardiac arrest patients. *Hua et al. (2009)* revealed that, in China, the survival rate of OHCA is less than 1% compared with 12% in the USA. Since 2010, continuous improvement of the survival rate in both in-hospital and out-of-hospital cardiac arrests has been demanded, and cardiopulmonary resuscitation (CPR), especially high-quality cardiopulmonary resuscitation (HQ-CPR), was critically important to patients and proved to save lives associated with good neurological outcomes (*Meaney et al., 2013*).

The conventional instructor-led basic support life (BLS) course requires learners to attend an in-person class annually or every 2 years to obtain certification. However, certain limitations for the conventional BLS training have been identified, including the course implementations are often without objective quantitative measurement of the quality of CPR, too much time consumption when mass delivery, and labor intensive with poor long-term skills retention and knowledge, etc. Insufficient exposure to hands-on practice results in trainees unfamiliar with the psychomotor CPR skills (*Braslow et al., 1997*). CPR skills decay rapidly within short times, studies showed the skills were able to be retained for only 3–6 months (*Meaney et al., 2012*; *Binkhorst et al., 2018*; *Cheng et al., 2018*; *Halm & Crespo, 2018*). In addition, unexpected psychological barriers and distractions may arise in an unfamiliar environment (*Liberman et al., 2000*). Therefore, novel pedagogics based on educational theories might help trainees to improve their learning and master of CPR skills.

The International Liaison Committee on Resuscitation (ILCOR) 2015 and 2020 international consensus on resuscitation recommended that high-frequency training improves the quality of CPR delivery and the confidence of practitioners, and individuals that are likely to encounter cardiac arrest (*e.g.*, emergency physicians and nurses), should be considered receiving BLS refresh training with a higher frequency (*Finn et al., 2015*; *Greif et al., 2020*).

*Yan et al. (2020)*, reported a high willingness to attend cardiopulmonary resuscitation training and encouraged making efforts to optimize and standardize a national model of CPR delivery training. Chinese national consensus on quality control of IHCA adult cardiopulmonary resuscitation suggested implementing high-frequency training for reducing skills decay as a prevailing existence among the healthcare providers (*Wang, Sun & Yangyang, 2018*). Meanwhile, *Chinese Medical Association Science Popularization Branch (2018)* emphasis on physicians and nurses with a high frequency of implementing CPR in clinical practice cannot benefit enough from the conventional course.

The Resuscitation Quality Improvement (RQI) program was launched by the American Heart Association (AHA) since 2015. It is an educational CPR training system combining self-adaptive learning, real-time feedback training, and spaced training (*Kuyt et al., 2021*).

The RQI program, unlike conventional BLS course, provides a high-reliable platform for simulation-based mastery learning and deliberate practice through low-dose, high-frequency quality improvement (*Dudzik et al., 2019*; *Panchal et al., 2020*). The RQI program is delivered in a mastery learning model theory, with both knowledge and skill baseline assessments conducted as the pretests, while each core skill training is delivered to allow the deliberate practice of learners, retraining of the skills and further repetition if needed (not passing certain high-achieving standards), and permitted to move to the next session only when achieving high standards of the post-assessment of the current learning unit.

Although it has been proved that high-frequency low dosage training is better than conventional training in CPR core skills learning, when compared with the accumulating data internationally, rare reporting of studies on the effect and quality improvement of CPR skills of RQI training program has been a challenge for its prevalence in China. We sought to deterimine whether effectiveness could be confirmed among Chinese trainees. Before 2021, Chinese clinicians were trained by a conventional training course for over 10 years and accumulated a massive population with BLS learning experience rather than RQI program. It was not clear whether RQI program was effective for both BLS initial certification and renewal training. Most of the previous studies (*Dudzik et al., 2019*; *Panchal et al., 2020*; *Kuyt et al., 2021*) examining the effectiveness of RQI were conducted in developed country. It is not clear whether the implementation of the RQI program was feasible and acceptable in a developing country with a different healthcare system (*e.g.*, China). Is it possible for Chinese trainees to accommodate this innovative training modality? Therefore, we would like to investigate the implementation of RQI program in the Chinese setting. At the beginning of the study design, we realized that there may be differences in CPR performance between novices and experienced providers who have participated in conventional BLS training, as well as differences in confidence in CPR skills and satisfaction with RQI procedures. So we designed this study, with the following aims: 1. to compare the performance of CPR core technique between novice performance and trainees experienced in conventional BLS training post-RQI entry course; 2. to describe how the Chinese participants' satisfaction with the RQI training program and whether the self-confidence on performing CPR could be improved.

## MATERIALS AND METHODS

### Population

This is a single-center observational study conducted from May 9, 2021 to June 25, 2021, in the emergency department of a tertiary hospital located in Beijing. Informed consent was received from all eligible participants and could withdraw at any stage of the study. Participants including both physicians and nurses who have previously received conventional BLS training were defined as the BLS group and those who did not participate in the AHA BLS training as the Non-BLS group.

Inclusion criteria: 1. Physical health be able to perform chest compressions and bag-valve-mask ventilation maneuvers. 2. Emergency physicians and nurses, as well as

clinicians who rotate to emergency departments from other divisions during the period were eligible to be enrolled in the study.

Exclusion criteria: 1. Those physically unable to complete the course, *e.g.*, pregnancy. 2. Those who chose to exit the study cohort.

This study was approved by the Internal Review Board (IRB) of Peking Union Medical College Hospital (ZS-2877).

## RQI training

Based on high-frequency low dosage training, the learning algorithm of the RQI program was conducted in five secessions, the baseline assessment and four re-training quarters (*Cheng et al., 2018*). Details of the RQI program were described in the Supplementary Document. The RQI system consists of a cart (including adult/infant-sized manikins, footrests, electronic training systems and a laptop). Training session ranges from tens of minutes to one hour. Trainees were allowed to finish the training independently with the cart located in the working area of emergency department. The training certification can only be obtained by passing the knowledge and skill learning.

Performance data were collected through online learning and training management systems during courses. Under skill training, the program included two core techniques, which are chest compression and bag-valve-mask ventilation on adult- and infant-sized mannequins respectively. The baseline of each core technique would be obtained without any feedback. Outcome variables include key parameters of compression and ventilation: compression depth, compression rate, recoil depth, ventilation rate, ventilating volume, and the scores for each session. The score ranges from 0 up to 100, with calculation implemented by a proprietary weighting algorithm based on AHA recommendations. Trainees were required to achieve a score over 75 in each core technique practice before advancing to the next session of core skills training.

Area9 Lyceum is a cloud-based educational platform to provide online learning modules for schoolchildren, students and professionals. The entry course consists of baseline assessment and training of online adaptive learning for knowledge about the protocol of BLS by globally-leading Area9 Rhapsode™ learning and publishing platform, which is supported by Area9 Lyceum, followed by core technique practice (Available from: https://rqipartners.com/solutions/in-hospital-solutions/rqi-resuscitation-quality-improvement. Accessed 15 September 2022). If the participant fails, the platform would be programmed to provide individualized feedback so that the participant can remedy and complete the exercise; unlimited attempts are allowed until a passing score is reached (*Donoghue et al., 2021*). Psychomotor performance data would be recorded until passing all the practical sessions following the knowledge learning was completed and thereby a certification would be issued. The measurement of post-RQI performance was conducted with real-time feedback. Meanwhile, the number of attempts to obtain the certification was recorded.

## Survey

We designed an online survey using Jinshuju (www.jinshuju.net) platform to query participants' learning experience with the RQI program, to obtain their self-confidence and satisfaction with this novice training program. Questions were formulated based on training modalities, a review of the existing literature, and discussions with emergency clinicians and educating providers. To ensure the validity, we selected three senior emergency physicians to review each question for relevance and clarity. Questions were pilot tested among clinicians based in a major tertiary care hospital in Beijing and subsequently honed. For the survey questions, we conducted a pilot test among clinicians in our hospital and then honed it. Questionnaires consisted of five items with single-choice, multiple-choice, and open-ended questions asking about demographic characteristics, participants' previous BLS training experiences, satisfaction rating scores for RQI training program, the self-confidence of core technique implementation and their suggestions for future RQI training facilitating. A rating scale had been used for confidence and satisfaction evaluation. The rating ranges from 0–10 to a maximum of 10 points for confidence and satisfaction evaluation.

## Statistical analysis

Data were analyzed in GraphPad Prism software (version 8.0.2.263) and R v3.6.0 (R Core Team, 2019). Standard descriptive statistics were calculated as appropriate for the distribution of each variable. Descriptive statistics were used to describe the demographic characteristics of the two groups' participants; the Continuous parametric variables were compared with the T-test or the non-parametric Wilcoxon rank sum test, and categorical variables were compared by using the chi-square test. Levene's robust test (F test) was used to assess the equality of variance in continuous measures of CPR quality at baseline and post-RQI. For the analysis on attempts of core skill practice to achieve the training, the data was not normally distributed, Wilcoxon rank sum tests was used. Spearman's correlation coefficient was calculated for the correlation between CPR performance and self-confidence. Statistical significance was accepted at $P < 0.05$.

# RESULTS

## Demographic data

A total of 150 participants were enrolled in this study, with one participant dropped out due to pregnancy. One hundred and forty nine participants were included in the analysis. All subjects included 119 females (79.9%), and there are 45 doctors (30.2%) and 104 nurses (69.8%), respectively. One hundred and three (69.1%) out of all the 149 participants had previous experience of conventional BLS training. Compared with Non-BLS group, BLS group had statistical differences in age ($P = 0.009$), profession ($P = 0.001$), and working experience ($P = 0.007$). Other demographic characteristics did not show differences between the two groups. The time of last conventional BLS training is shorter than 3 months accounted for 60.2% of participants, 3–6 months accounted for 5.8%, 6–9 months accounted for 4.9%, 9–12 months accounted for 4.9%, Over 12 months accounted for 24.3%. More details showed in Table 1.

**Table 1 Comparison of basic information of all RQI trainees.**

| | Non-BLS (n = 46) | BLS (n = 103) |
|---|---|---|
| Gender, n (%) | | |
| Female | 38 (82.6) | 81 (78.6) |
| Male | 8(17.4) | 22 (21.4) |
| Age (year), n (%) | | |
| 18–29 | 20 (43.5) | 22 (21.4) |
| 30–39 | 21 (45.7) | 48 (46.6) |
| 40–49 | 5 (10.9) | 27 (26.2) |
| 50–59 | 0 (0.0) | 6 (5.8) |
| Height (m), n (%) | | |
| 1.5–1.59 | 10 (21.7) | 11 (10.7) |
| 1.6–1.69 | 24 (52.2) | 65 (63.1) |
| 1.7–1.79 | 8 (17.4) | 20 (19.4) |
| 1.8–1.89 | 4 (8.7) | 7 (6.8) |
| Weight (kg), n (%) | | |
| 40–50 | 6 (13.0) | 13 (12.6) |
| 51–60 | 22 (47.8) | 42 (40.8) |
| 61–70 | 8 (17.4) | 30 (29.1) |
| 71–80 | 6 (13.0) | 8 (7.8) |
| 81–90 | 1 (2.2) | 7 (6.8) |
| 91–100 | 2 (4.3) | 3 (2.9) |
| >100 | 1 (2.2) | 0 (0.0) |
| Occupation, n (%) | | |
| Physician | 13 (28.3) | 32 (31.1) |
| Nurse | 33 (71.7) | 71(68.9) |
| Working years (year), n (%) | | |
| <5 | 20 (43.5) | 19 (18.4) |
| 5–9 | 11 (23.9) | 20 (19.4) |
| 10–14 | 6 (13.0) | 20 (19.4) |
| 15–19 | 5 (10.9) | 16 (15.5) |
| ≥20 | 4 (8.7) | 28 (27.2) |
| Degree obtained, n (%) | | |
| Medical student | 5 (10.9) | 17 (16.5) |
| Bachelor's | 30 (65.2) | 56 (54.4) |
| Master's | 7 (15.2) | 15 (14.6) |
| Doctoral | 4 (8.6) | 15 (14.6) |
| Working area, n (%) | | |
| Emergency room | 15 (32.6) | 27 (26.2) |
| Triage area | 3 (6.5) | 10 (9.7) |
| Emergency intensive care unit | 10 (21.7) | 8 (7.8) |
| General ward | 4 (8.7) | 9 (8.7) |
| Observing room | 4 (8.7) | 16 (15.5) |

| Table 1 (continued) | | |
|---|---|---|
| | **Non-BLS (*n* = 46)** | **BLS (*n* = 103)** |
| Infusion room | 5 (10.9) | 14 (13.6) |
| Other | 5 (10.9) | 19 (18.4) |
| Time to previous conventional BLS course (month), *n* (%) | | |
| <3 | 0 (0) | 62 (60.2) |
| 3–5 | 0 (0) | 6 (5.8) |
| 6–9 | 0 (0) | 5 (4.9) |
| 10–12 | 0 (0) | 5 (4.9) |
| >12 | 0 (0) | 25 (24.3) |

## CPR performance between non-BLS and BLS groups

Among the baseline of RQI core skills, overall achievements of the goals for AHA recommendation of core skills were compression and ventilation (*Cheng et al., 2020*).

For chest compression, there was no statistically significant difference in mean value of total adult scores (65.96 in Non-BLS group and 54.98 in BLS group, $P = 0.069$) and total infant compression scores (46.67 in Non-BLS group and 36.36 in BLS group, $P = 0.108$). The mean value of total ventilation score on the adult-sized mannequin was higher in Non-BLS group compared to BLS group (58.33 *vs* 41.3, $P = 0.016$). There was no statistically significant difference in mean value of total infant ventilation scores (39.72 in Non-BLS group *vs* 34.13 in BLS group, $P = 0.336$). Between the two groups, under compression skill activity, adult compression rate was faster in BLS group compared to that in the Non-BLS group (123.88 per min *vs* 113.93 per min, $P = 0.022$), and out of recommended compression rate range of the guidelines. For ventilation, participants in the BLS group had a faster ventilation rate for adults (21.1 per min in BLS group *vs* 15.13 per min in Non-BLS group, $P = 0.006$) and infants (32.73 per min in BLS group *vs* 26.04 per min in Non-BLS group, $P = 0.027$) than those in the Non-BLS group.

Among the post RQI performance, the parameters of compression practice listed in Table 2, each group past the training, achieved the standard goal of compression rate, depth and recoil rate, and there were no statistically significant differences between the two groups in compression performance. Under ventilation, all of the participants obtained 75 or more scores, and the performance of mean infant ventilation time (385.5 ms *vs* 439.15 ms, $P = 0.006$) on psychomotor performance was statistically different between the two groups. The number of attempts to achieve certificated core technique practice was lower in BLS group on both adult and infant mannikins in the post RQI group than the other group ($P < 0.05$). (Table 2).

## SURVEY

### Self-confidence evaluation

Trainees in BLS group had better confidence in the performance of adult chest compressions and ventilation ($P < 0.01$). Performance of adult ventilation had the highest rating of self-confidence in all participants. However, the performance of infant ventilation

**Table 2 Comparison of CPR quality between non-BLS group and BLS group.**

| | Adult/ Infant | Baseline Non-BLS group (n = 46) | BLS group (n = 103) | P value | Post RQI Non-BLS group (n = 46) | BLS group n = 103 | P value |
|---|---|---|---|---|---|---|---|
| Compression rate (min$^{-1}$), mean (SD) | Adult | 113.93 (17.96) | 123.88 (26.48) | 0.022* | 108.43 (7.11) | 109.30 (8.27) | 0.539 |
| | Infant | 115.63 (33.00) | 128.12 (43.36) | 0.084 | 109.11 (7.14) | 109.58 (7.58) | 0.723 |
| Compression depth (mm), mean (SD) | Adult | 48.78 (9.45) | 48.00 (10.31) | 0.664 | 54.21 (3.40) | 54.25 (3.52) | 0.945 |
| | Infant | 34.86 (7.15) | 34.74 (6.63) | 0.926 | 39.44 (1.88) | 39.17 (1.62) | 0.361 |
| Compression release depth (mm), mean (SD) | Adult | 4.47 (2.12) | 5.02 (3.14) | 0.283 | 3.42 (1.97) | 3.08 (1.91) | 0.332 |
| | Infant | 2.49 (2.84) | 2.48 (2.45) | 0.983 | 1.54 (1.41) | 1.92 (1.52) | 0.149 |
| Compression score, mean (SD) | Adult | 65.96 (31.76) | 54.98 (34.53) | 0.069 | 94.70 (5.69) | 93.90 (5.96) | 0.448 |
| | Infant | 46.67 (36.22) | 36.36 (35.81) | 0.108 | 95.26 (5.17) | 93.95 (5.67) | 0.184 |
| Ventilation inflation time (ms), mean (SD) | Adult | 911.00 (241.50) | 839.02 (222.70) | 0.078 | 766.09 (151.10) | 806.84 (208.31) | 0.235 |
| | Infant | 532.72 (178.06) | 514.49 (172.46) | 0.556 | 385.50 (101.34) | 439.15 (110.48) | 0.006* |
| Ventilation volume (ml), mean (SD) | Adult | 535.74 (124.03) | 538.72 (158.57) | 0.910 | 532.13 (55.37) | 534.26 (57.89) | 0.834 |
| | Infant | 43.59 (15.28) | 40.83 (15.69) | 0.320 | 30.76 (4.26) | 31.78 (3.81) | 0.149 |
| Ventilation rate (min$^{-1}$), mean (SD) | Adult | 15.13 (6.49) | 21.10 (13.74) | 0.006* | 11.93 (1.44) | 12.19 (1.08) | 0.224 |
| | Infant | 26.04 (10.90) | 32.73 (18.90) | 0.027* | 26.15 (3.22) | 26.07 (3.24) | 0.884 |
| Ventilation score, mean (SD) | Adult | 58.33 (38.09) | 41.30 (39.61) | 0.016* | 95.20 (6.52) | 95.85 (6.44) | 0.566 |
| | Infant | 39.72 (31.67) | 34.13 (33.03) | 0.336 | 93.52 (7.87) | 94.33 (6.68) | 0.520 |
| Times of attempt to pass for Compression, median (IQR) | Adult | 1.0 | 1.0 | NA | 1.0 (1.0, 2.0) | 1.0 (1.0, 1.0) | 0.028* |
| | Infant | 1.0 | 1.0 | NA | 1.0 (1.0, 2.0) | 1.0 (1.0, 1.0) | 0.027* |
| Times of attempt to pass for ventilation, median (IQR) | Adult | 1.0 | 1.0 | NA | 1.0 (1.0, 2.0) | 1.0 (1.0, 1.0) | 0.034* |
| | Infant | 1.0 | 1.0 | NA | 1.0 (1.0, 2.0) | 1.0 (1.0, 1.0) | 0.038* |

Notes:
* Statistically significant difference.
RQI, resuscitation quality improvement; BLS, basic life support; SD: standard deviation; IQR, interquartile range; NA, not available.

**Table 3 Self-confidence rating on CPR post RQI entry course.**

| | Non-BLS group | BLS group | P value |
|---|---|---|---|
| Adult compression | 8.74 (1.39) | 9.44 (0.99) | 0.001* |
| Infant compression | 8.26 (1.41) | 9.00 (1.48) | 0.004* |
| Adult ventilation | 9.11 (1.19) | 9.67 (0.63) | <0.001* |
| Infant ventilation | 8.57 (1.31) | 9.17 (1.38) | 0.012* |

Notes:
* Statistically significant difference.
BLS, basic life support.

gained the lowest rating (Table 3). To explore the correlation between confidence and performance, Fig. 1 demonstrates no significant relationship between CPR performance (adult compression, adult ventilation, infant compression, and adult ventilation) after RQI training and their self-confidence.

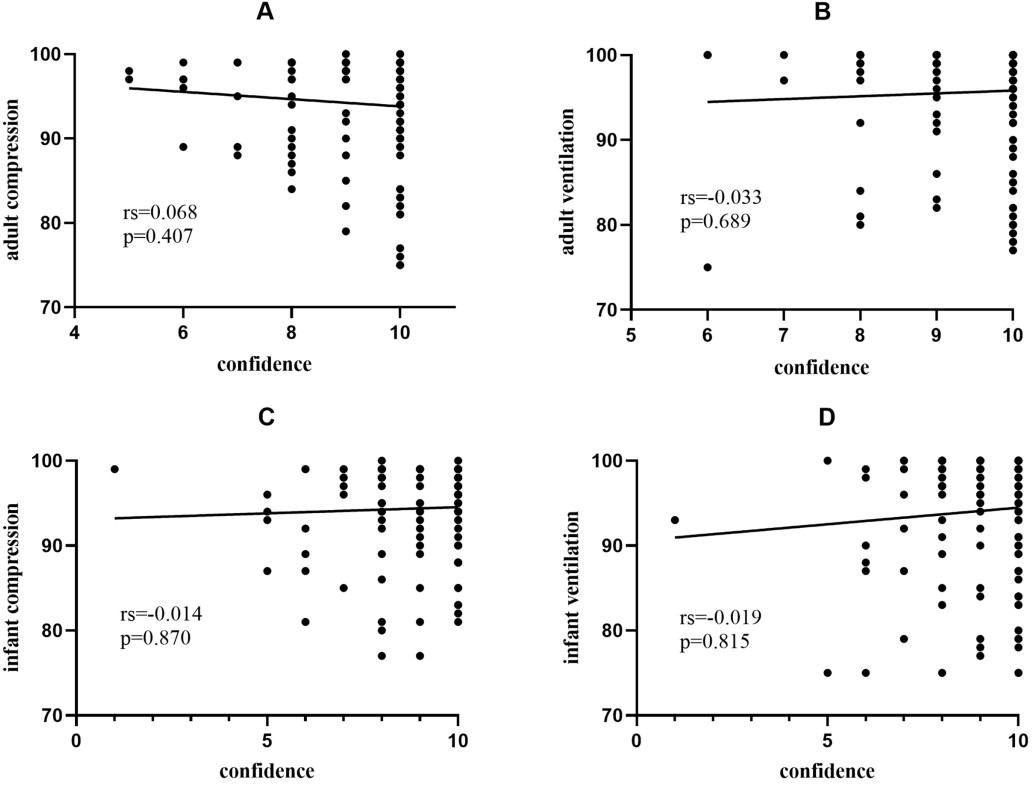

**Figure 1** **Correlation between compression/ventilation scores (post-RQI training) and self-confidence scores.** (A–D) Showed no significant correlation between CPR performance (post-RQI training) and the operators' self-confidence.

## Trainee perception regarding RQI program

For both groups, trainees demonstrated similar attitudes toward RQI program facilitating (Table 4). BLS group showed high ratings on the statement of organization of simulation cart (89.3%, strongly agreed). Nevertheless, for Non-BLS, real-time feedback during CPR acquired the highest rating (84.8, Strongly agreed). Both groups had the lowest satisfaction with no instructor-facilitated training (72.8% in BLS group *vs* 65.2% in No-BLS group, strongly agreed). The proportion of satisfaction ratings regarding RQI training program with full marks were respectively 78.6% and 69.6% in BLS group and another.

## DISCUSSION

Previous studies have reported on the excellent performance of RQI indicated the successful improvement of healthcare providers' CPR psychomotor competence and confidence/satisfaction in different hospital settings (*Kuyt et al., 2021*; *Dudzik et al., 2019*). Our study enrolled 149 participants certificated in the entry course and all of the participants improved their performance in all of the core techniques. All of the participants achieved the training and be certificated, both groups showed high satisfaction with RQI entry course, which is consistent with those previous studies. Two strategies were introduced to improve the compliance of RQI program, that trainees were allocated in small groups and a team leader was set in each group to supervise the training.

**Table 4 Results of RQI satisfaction survey.**

| | BLS group | Strongly agreed (%) | Agreed (%) | Neutral (%) | Disagreed (%) | Strongly disagreed (%) | P value |
|---|---|---|---|---|---|---|---|
| I prefer the e-learning of the RQI program to classroom teaching. | No | 38 (82.6) | 8 (17.4) | 0 (0.0) | 0 (0.0) | 0 (0.0) | 0.226 |
| | Yes | 90 (87.4) | 10 (9.7) | 3 (2.9) | 0 (0.0) | 0 (0.0) | |
| I like the module of virtual simulation for team resuscitation. | No | 37 (80.4) | 9 (19.6) | 0 (0.0) | 0 (0.0) | 0 (0.0) | 0.280 |
| | Yes | 91 (88.3) | 11 (10.7) | 1 (1.0) | 0 (0.0) | 0 (0.0) | |
| I like the form of cart simulator. | No | 37 (80.4) | 9 (19.6) | 0 (0.0) | 0 (0.0) | 0 (0.0) | 0.226 |
| | Yes | 92 (89.3) | 11 (10.7) | 0 (0.0) | 0 (0.0) | 0 (0.0) | |
| I like the form of control my free time in the psychomotor skill. | No | 38 (82.6) | 8 (17.4) | 0 (0.0) | 0 (0.0) | 0 (0.0) | 0.604 |
| | Yes | 90 (87.4) | 13 (12.6) | 0 (0.0) | 0 (0.0) | 0 (0.0) | |
| I think the allocation of time of the simulation part in the skill learning is reasonable. | No | 38 (82.6) | 8 (17.4) | 0 (0.0) | 0 (0.0) | 0 (0.0) | 0.845 |
| | Yes | 88 (85.4) | 15 (14.6) | 0 (0.0) | 0 (0.0) | 0 (0.0) | |
| I think the training environment of simulator is good. | No | 37 (80.4) | 9 (19.6) | 0 (0.0) | 0 (0.0) | 0 (0.0) | 0.139 |
| | Yes | 90 (87.4) | 10 (9.7) | 3 (2.9) | 0 (0.0) | 0 (0.0) | |
| I think the real-time feedback of the RQI program is very helpful. | No | 39 (84.8) | 7 (15.2) | 0 (0.0) | 0 (0.0) | 0 (0.0) | 0.776 |
| | Yes | 88 (85.4) | 14 (13.6) | 1 (1.0) | 0 (0.0) | 0 (0.0) | |
| I think the content settings of the RQI program are good. | No | 36 (78.3) | 10 (21.7) | 0 (0.0) | 0 (0.0) | 0 (0.0) | 0.377 |
| | Yes | 88 (85.4) | 14 (13.6) | 1 (1.0) | 0 (0.0) | 0 (0.0) | |
| I think no instructor-led training is more conducive to my learning | No | 30 (65.2) | 9 (19.6) | 6 (13.0) | 0 (0.0) | 1 (2.2) | 0.629 |
| | Yes | 75 (72.8) | 13 (12.6) | 12 (11.7) | 2 (1.9) | 1 (1.0) | |

**Note:**
RQI, resuscitation quality improvement; BLS, basic life support.

Compared to the Non-BLS group, participants who had conventional instructor-led training experience had fewer attempts on achieving certification of core techniques. BLS group showed a higher confidence rating for both compression and ventilation compared to the Non-BLS group on adult CPR performance ($P$ value < 0.05). The RQI training program had been well accepted in the group of BLS (fewer attempts and time consumption on core skills training, $P$ value < 0.05 for compression and ventilation respectively). Trainees in our study achieved a qualified CPR performance in the RQI training as all trainees passed the training assessment. After RQI training, trainees in both groups demonstrated improvement among each core technique compared with baseline, most of the compression and ventilation outcomes have been significantly improved. With psychomotor CPR skills training, RQI program followed simulation-based mastery learning, facilitated program by deliberate practice to guarantee both novices and trainees with previous training experience meet the high quality of core skills performance. Conventional instructor-led CPR training has been proved to produce higher consumption in human resources, additional spending on training space renting and alternative demands for healthcare providers impacting on time of patient care. On the contrary, RQI program showed advantages compared to conventional BLS training in aspects of its lower cost, no instructor assistance, and real-time feedback (*Dudzik et al., 2019*; *Oermann et al., 2020*; *Panchal et al., 2020*; *Schmitz et al., 2021*; *Kuyt et al., 2021*; *Donoghue et al., 2021*). Furthermore, we found there was no significant correlation

between the compression/ventilation score after RQI training and the self-confidence score. The medical staff may feel good about their self-perceptions of CPR quality, but they are often mismatched with the actual skills (*Cheng et al., 2019*; *Troy et al., 2019*). This further points to the need to use real-time feedback in CPR training.

The participants in BLS group are more senior, had longer careers, and showed abundant clinical experience in this study. However, these findings did not indicate superior CPR performance outcomes. Up to 60.2% of learners took participant in the BLS training according to the latest CPR guidelines within 3 months in this study, adult compression rate was faster in BLS group, and faster adult and infant ventilation rates than those in Non-BLS group. *Kandasamy et al. (2019)* also found that participants in the BLS training group were 2.5 times more likely to be over-compressed than the lay rescuers' cohort group. Three explanations for the results of excessive compression and ventilation were: 1. The previous training emphasized the negative correlation between lower rates and the impact on the implementation of compression (*Kandasamy et al., 2019*). Compression rate during CPR had been corrected from over 100 bpm to 100–120 bpm since 2015 CPR guidelines (*Travers et al., 2015*). But the effect of emphasizing rapid compression at CPR may still exist. 2. Conventional BLS training seems to lack a reliable and efficient objective assessment. The quality of CPR in BLS course depends on the instructor's subjective visual assessment (*Cheng et al., 2015*; *Jones et al., 2015*). visual assessment by instructors was found to be less reliable for CPR skills performance assessment after the course. 3. Studies have also shown that BLS skills will experience a decline over time, which can occur in 3–6 months post-training (*Meaney et al., 2012*; *Binkhorst et al., 2018*; *Cheng et al., 2018*; *Halm & Crespo, 2018*). The RQI program, with distributed practice, mastery learning and real-time feedback, could compensate for the deficiency of conventional BLS training programs. *Lin et al. (2018)* found that compare to the conventional BLS course, the distributed CPR training with real-time feedback in pediatric healthcare providers was able to improve chest compression metrics (depth, rate and recoil) at 3 months with retention of CPR skills throughout the study. In this study, we found that trainees in BLS group who went through RQI training required fewer attempts to pass the course, showing a steeper learning curve, and a faster re-mastery of CPR skills during retraining, reducing the total time spent on retraining. This result could be a rational explanation for conducting RQI program among the population with previous conventional instructor-led training representing most of the healthcare providers in Mainland China.

RQI not only provide excellent CPR skills, but also established confidence for novice and experienced learners. Although the level of confidence before the RQI program training was not evaluated, the BLS group demonstrated better confidence post-training. This might be caused by the trainees who experienced both BLS training and also had more clinical practice to enhance their confidence than novices. However, in this study exposed trainees with previous learning experience had poorer performance of core techniques during baseline assessment, indicating that confidence might be a false trust in previously learned skills (*Hopstock, 2008*). Mismatching of "false self-confidence" and poor performance may lead to irrational compression, as well as ventilation, during CPR.

Subsequent to this situation, improper clinical practice may affect the quality of patient care. Therefore, the emphasis on CPR training with mastery learning was important.

This study found that almost all participants are highly satisfied with the RQI program, indicating that it had a good acceptance and a subjective program success. These findings are congruent with similar research conducted in the USA by *Dudzik et al. (2019)*, in which the RQI had higher satisfaction and increased confidence in CPR skills for health care providers at a community hospital. They found that 89 (67.4%) out of the 132 respondents agreed or strongly agreed that RQI was their preferred BLS training method compared with the traditional model or online HeartCode BLS options. Our findings demonstrated the highest satisfaction levels for RQI among respondents based on the unique features of the program, such as the form of a cart simulator in the BLS group and real-time feedback in the Non-BLS group. Trainees with previous experience of conventional BLS training were familiar with the process of training, and their attention was more likely to be drawn to the newer modality of the cart. The feedback system during CPR training can guide the real-time correction of CPR performance and can improve compliance of high quality. There are similar researches on non-conventional BLS training methods involving online eSimulation and real-time audiovisual manikin feedback technology (*Yeung et al., 2009*; *Lin et al., 2018*; *Weston et al., 2019*).

"No instructor-involvement during training" is one of the characteristics of the RQI program, but interestingly, among all of the participants' ratings, such a feature received relatively lower satisfaction. The purpose of conventional training with instructor-involved training was targeting for assessing learners' CPR skills and correcting errors to ensure high-quality performance. However, As we mentioned before, the instructor may not always be able to identify the appropriate chest compression depth and rescue breaths without real-time feedback (*Hansen et al., 2019*). Previous studies revealed no differences in terms of effectiveness, when comparing self-learning *vs* instructor-led teaching (*Mpotos et al., 2011*, *2012*; *Mardegan, Schofield & Murphy, 2015*). Two trials found that self-learning is not inferior to instructor-led learning in the short term. However, trainees keep better performance of CPR practical skills for instructor-led learning after BLS training (*Pedersen et al., 2018*; *Bylow et al., 2019*). As the current model of BLS training is mainly dominated by classroom-based instructor-involved training in our country (*Finn et al., 2015*; *Cheng et al., 2020*; *Wang, Meng & Yu, 2018*), and RQI program may be one of the effective alternative courses.

Under the post-pandemic era, BLS training has been a challenge to Chinese clinicians, instructor led and classroom gathering has been restricted conducted in training site, calling for other training programs instead of conventional course for better implementation in China (*Yan et al., 2020*). It has been reported that the success rate for BLS among outpatients was significate lower than the data of the USA, and Chinese clinicians have been making a great effort on improving it (*Wang, Meng & Yu, 2018*; *Wang, Sun & Yangyang, 2018*). With a massive population having been training with a conventional course, it was equally important for us to explore the perception and effectiveness of training for both novice and experienced healthcare providers, because skill retention was a key content for all populations. In our study, we found the trainees with

previous learning experience showed better improvement among the core skills practice and fewer numbers of attempts for passing the training with lower baseline scores compared with novice, they also reported better confidence aligned with training outcomes, which could be explained that for those with learning experience RQI program would be another option for BLS training, but the feasibility of this program should be discussed until it has been facilitated for one year considered other data should be involved into the measurements.

### Strengths and limitations of the study

To our knowledge, this is the first time that the RQI program has been studied and reported in mainland China. Although there were some studies about real-time feedback of chest compressions in our country, ventilatory feedback is rarely involved (*Jiang et al., 2010*; *Zhou et al., 2020*; *Meng et al., 2021*). We assessed participants perspective of RQI program training which encourage further training and research to prove the feasibility of prevalence in China. In addition, there is another strength in our study to compare the performance of CPR core technique between novice performance and trainees experienced in conventional BLS training post RQI entry course.

### Limitations

There are several limitations of this study. Firstly, this study is an observational study from a single center in China. Due to time constraints, continuous testing and cost-benefit analysis have not been carried out, and these studies will be carried out in the future. Secondly, this research does not directly compare RQI with instructor-led BLS course, and it is difficult to explain which is better, but we believe that the two are complementary to each other. Thirdly, as there was no satisfaction survey results obtained from trainees before the RQI study, it would be difficult to explain the changes in satisfaction and self-confidence in the current study. The evaluation after training still supports the positive evaluation of RQI to a degree. Lastly, considering the present study and evaluation of the effectiveness of BLS training for medical providers, the Kirkpatrick model was used. It is an evaluation methods with four levels (*Curran & Fleet, 2005*; *Dorri, Akbari & Dorri Sedeh, 2016*). We only presented the first two levels (learner reaction and learning). The follow-up may need to be expanded to a larger group of people.

## CONCLUSIONS

The RQI program can provide excellent CPR performance skills for participants. For trainees with previous experience in BLS training, the RQI program certificate the effectiveness of training while reducing the number of attempt for past the training, they also demonstrated higher confidence compared to the novice. For all participants, the RQI program showed a high rating of satisfaction. We might consider that RQI program could be an option for the prevalence of BLS training in mainland China, with broadly training of a conventional BLS course, but needed a large study to confirm.

## ACKNOWLEDGEMENTS

The authors would like to thank the resuscitation team of Emergency Department of the PUMCH for preliminary work related to this program.

### Funding

This work was supported by the National Center for Health Professions Education Development Funding Project for Simulated Medical Education Research (No. 2021MNYB02), and the Chinese Academy of Medical Sciences and the Peking Union Medical College Education Funding Project for the Construction of First-class Discipline (No. 2021zlgc1101), and Medical Education Scholar Program, Chinese Academy of Medical Sciences and Peking Union Medical College Education Funding Project (No. 2022zlgc0712). The funders had no role in study design, data collection and analysis, decision to publish, or preparation of the manuscript.

### Grant Disclosures

The following grant information was disclosed by the authors:
National Center for Health Professions Education Development Funding Project for Simulated Medical Education Research: 2021MNYB02.
Chinese Academy of Medical Sciences and Peking Union Medical College Education Funding Project for the Construction of First-class Discipline: 2021zlgc1101.
Medical Education Scholar Program, Chinese Academy of Medical Sciences and Peking Union Medical College Education Funding Project: 2022zlgc0712.

### Competing Interests

The authors declare that they have no competing interests.

### Author Contributions

- Hui Jiang conceived and designed the experiments, analyzed the data, prepared figures and/or tables, and approved the final draft.
- Liang Zong conceived and designed the experiments, performed the experiments, analyzed the data, prepared figures and/or tables, and approved the final draft.
- Fan Li performed the experiments, authored or reviewed drafts of the article, and approved the final draft.
- Jian Gao performed the experiments, authored or reviewed drafts of the article, and approved the final draft.
- Huadong Zhu conceived and designed the experiments, performed the experiments, authored or reviewed drafts of the article, and approved the final draft.
- Di Shi conceived and designed the experiments, performed the experiments, analyzed the data, prepared figures and/or tables, authored or reviewed drafts of the article, and approved the final draft.

- Jihai Liu conceived and designed the experiments, performed the experiments, authored or reviewed drafts of the article, and approved the final draft.

## Ethics

The following information was supplied relating to ethical approvals (*i.e.*, approving body and any reference numbers):

The Institutional Review Board of Peking Union Medical College Hospital approved this study. (Ethical Application Ref: ZS-2877).

## Data Availability

The raw measurements are available in the Supplemental File.

## Supplemental Information

Supplemental information for this article can be found online at http://dx.doi.org/10.7717/peerj.14345#supplemental-information.

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
