# Peer review of "Initial implementation of the resuscitation quality improvement program in emergency department of a teaching hospital in China"

_PeerJ, doi:10.7717/peerj.14345_

## Round 0.1 · original submission · Major Revisions

Thank you for submitting your manuscript for review. The content and contest of the manuscript is of interest and contributes to the resuscitation literature. Should you wish to revise and resubmit your manuscript please pay careful attention to the reviewers' recommendations, particularly reviewer #2's detailed and thoughtful comments.

Reviewer 1 ·

Basic reporting

The manuscript by Jiang et al. presented an interesting study that looks into the early RQI implementation in China. As far as I am aware, the topic of the this manuscript is of great importance as the training of healthcare professionals on their resuscitation skills should be constantly emphasized. Interestingly, the study described an early phase of the RQI implementation in a Chinese hospital, which I believe is indeed first of its kind to touch on the subject.

The authors have made extensive background introduction on why such study is required, which I think is quite convincing, especially on why it is necessary to conduct the relevant study in the Chinese content and how that would be potentially different with the existing study on RQI and resuscitation training.

The written language of this manuscript is overall friendly for international readers, but some of the expressions do have room for improvement. I would suggest the authors to focus on carefully phrasing the contents in this article.

e.g. The last sentense of Para. 3 of the Introduction section describing RQI program is a mastery learning model, would require feature careful phrasing to acurately capture the features.

Experimental design

The current presentation of the manuscript is quite well structured, with enough data supporting the 1st research question (To compare the performance of CPR core technique between novice performance and trainees experienced in conventional BLS training post RQI entry course). The outcome meaasures used in the study is believed to be sufficient to answer the question.

However, it would be hard for me to be convinc on how well the 2nd research question (To explore the feasibility of RQI training in China based on the broadly existed conventional BLS training) has been answered. What measures could define "feasibility"? I personally don't feel confident with the RQI program being feasible to implement in China only by judging from trainee self-evaluation, satisfaction etc. I think falls under the field of implementation science, where trainee/learner acceptance is important, but not enough. Buy-ins from other stakeholders, like the hospital leadership etc. should also be considers.

If the authors would want to stick to a feasibility study of the RQI program implementation in China, I would suggest to include more data from other stakeholders. Nevertheless, the learner satisfaction itself is a critical first step towards further buy-ins from others.

Validity of the findings

It's interesting to see that the authors compared the impact of w/o prior BLS training experience on the learning outcomes of the RQI program, which is both quite new and also make sense. For the novelty, few existing publications actaully made comparison from this angle; for making sense, it's not unusual to have quite percentage of healthcare professionals to have already received BLS training. I think this aspect itself is a strength (of interestingness) and could be mentioned in the discussion part.

Again, I would want to see data on other aspects of buy-in/acceptance if the author want to stick to explaining on the feasibility. Alternatively, the authors could just focus on the learners' perceptions on the RQI program.

It would also be good to mention about the future directions of the study, as apparently this is still a on-going project (hopefully).

Reviewer 2 ·

Basic reporting

Thank you for inviting me to review the manuscript titled “Initial implementation and learner satisfaction of the resuscitation quality improvement program in the emergency department of a tertiary hospital in China”. The authors reported the learning outcomes (i.e. CPR qualities), self-confidence, and satisfaction with the RQI training curriculum in frontline acute care providers in tertiary care centers in Beijing, China. The manuscript has some merits, however, I could not recommend the manuscript proceed to publication, until substantial revisions are made.

I found the manuscript difficult to read. There are numerous grammatical issues. I suggest the authors seek a professional language editing service before resubmission. I will not mention the grammatical issues for the rest of this review.

Title: Only satisfaction was mentioned in the title. However, outcome measures of the study included CPR performance, self-confidence, and satisfaction. The title did not seem to align with the rest of the manuscript.

Abstract: The results section in the abstract was a bit too long and off focus. I encourage the authors to carefully think about the key messages they wish to deliver in this manuscript. The key messages might not necessarily be the significant results.

Introduction:
Line 90 – 93: More work is required to address the research gap on the topic. Why is it important to investigate the implementation of the RQI program in China?

Experimental design

It is not clear to me whether this work is hypothetical research or exploratory research? If former, what is the main hypothesis? If latter, what is explored and why is that important? Specifically, why is it important to compare novices with experienced providers?

Aims:
Line 94 – 95: Why did the author compare novice with experienced healthcare providers? What is the implication of this comparison?
Line 96 – 97: The 2nd objective is difficult to achieve without data from the other stakeholders. I suggest the authors revise the 2nd aim. Please consider: “(2) To describe the self-confidence on performing CPR and satisfaction to the training in participants”

Methods
Line 129: What is Area9 Lyceum? Please clarify.
RQI training: A more detailed RQI training curriculum is recommended. Include a document as supplementary online content if needed.

Validity of the findings

I am overall satisfied with the validity of the finding. However, a few important outcome measures were missing.

RQI outcome measure: please indicate how was the post-RQI performance measured? Was there real-time feedback during the post-RQI assessment? A few important CPR metrics were missing in the outcomes, such as the percentage of guideline-compliant compression depth, rate, and recoil.

Survey: the process of the questionnaire development, validity, and reliability evidence were overall well described in the session.

Statistical analysis: I am not convinced that the number of attempts to pass the training (count data) was normally distributed. Non-parametric methods are recommended (i.e., Wilcoxon rank sum tests).

Results
Demographics (Table 1): Most of the readers outside China will not understand the Job title (primary, intermediate, deputy senior). I suggest the author either delete this variable or explain each item in the footnotes. For the variable Education: consider replacing “undergraduate” with “bachelor's degree” and replacing “Junior college” with “No degrees held”.
CPR performance: This section needs to be reorganized. I suggest one paragraph for baseline performance and another paragraph for post-RQI performance. Please report important outcomes instead of statistically significant outcomes.
Table 2: Please clarify what the numbers in the bracket stand for. Consider reporting the number of attempts to pass with median and interquartile range.
Self-confidence evaluation: Please indicate what was the highest self-confidence score participant could possibly report. It would significantly improve this manuscript if the authors could report the correlation between the self-confidence score and the compression/ventilation score. It is less of meaning if only reporting self-confidence alone, without linking it to the actual performance.
Please clarify the numbers in the brackets in Table 3.

Additional comments

Discussion:
Line 203 – 217: Please avoid repeating the results. Instead, summarize and paraphrase the results.
Most of the discussion focused on why and how RQI worked. I expected to see one paragraph discussing why it is important to implement RQI in China.
Strength (Line 294 – 300): this paragraph should not focus on the strength of the RQI program but on the strength of this research.

Conclusion:
Line 315: “RQI could correct skills decay with reducing the time cost”. The study did not examine skill decay, time, and cost.

---

## Round 0.2 · Minor Revisions

Thank you for your patience with the review process. Please see the requested remaining revisions from the two reviewers assigned to your manuscript.

Reviewer 1 ·

Basic reporting

There has been a significant improvement in this version of the manuscript. However, I feel that there are a few things the authors should further clarify.

1. The revised 2nd Research Aim is somewhat confusing. Should it be something similar to "(2) To describe how the Chinese participants react to the RQI training program in aspect to their satisfaction and whether the self-confidence on performing CPR could be improved." ?

2. Is the file "RQI supplement" reused with permission by the AHA. If not, then I would strongly suggest the authors describe the RQI program within the institution with their own words and figures to avoid any copyright issues.

Experimental design

no comment

Validity of the findings

Lines 320-323, I don't see the point of the discussing the skill retention issue as the authors have not been focusing on this aspect (according to authors' response towards Reviewer #2). If the authors were trying to address those trainees with prior BLS training experience would recover their skill (reaching certain skill standard) with shorter time than that without training experience, then the discussion should be rephrased to capture this point.

Reviewer 2 ·

Basic reporting

Thank you for inviting me to review this manuscript. I'm glad to see the manuscript has been improved a lot. The English language is much easier to follow compared to the previous version. However, there were quite a few language issues and gramma error in the manuscript. I have provided some examples (See additional comments for details). I suggest the authors check gramma sentence by sentence before resubmission.

Experimental design

The description of methods also improved significantly. The additional analysis was conducted and interpreted appropriately. A few minor errors need to fix. See additional comments for details

Validity of the findings

The results section with all tables and figures were appropriate. The statement of conclusion has been improved. There were some minor issues. See additional comments for details.

Additional comments

Introduction
The last paragraph of introduction is important and it needs extra work. This paragraph is supposed to address the research gap and the aims of the study.
Please consider the following gaps,
(1) It was not clear whether RQI program was effective for both BLS initial certification and renewal training.
(2) Most of the previous studies examining the effectiveness of RQI were conducted in developed country. It is not clear whether the implementation of RQI program was feasible and acceptable in a developing country with a different healthcare system (e.g. China)
The authors should clearly state the research gap and align them with the study aims.
A few other gramma issues below,
Line 94: “we could not stop considering that whether the effectiveness can be confirmed or not among Chinese trainees”. Gramma error
Line 97 – 99 “Will RQI … innovative training modality? “ Gramma error.
Line 102: Consider replace “veterans” with “experienced providers”

Material and Methods
Population: Please clarify in the section that the participants include both physicians and nurses
Line 143: The author did not explain what Area 9 lyceum is in the manuscript. Please add one or two sentences in the manuscript to explain.

Results
Line 179: If I understand correctly, 149 participants were included in the analysis. If that is the case,
“A total of 150 participants were enrolled in this study, with one participant dropped out due to pregnancy. One hundred and forty nine participants were included in the analysis.”
Table 1: I would remove the p-value column.
Line 191: Please consider changing “Under compression” into “For chest compression”
Line 101: for numbers in the brackets “(65.96 in Non-BLS…) Please clarify if the numbers are mean or median in the manuscript.
Line 198 Please consider present “min-1” as “per min”
Table 2: “Release Depth (cm)” Please double check the unit of this row. I believe it should be mm
Some numbers in the last 4 rows are missing.
Discussion
Line 251: “On the contrary, RQI program showed advantages… real-time feedback” Please check gramma.
Line 286: “RQI not only provide excellent…novice and re-trainer” Please check gramma. Please consider replace “re-trainer” to “experienced learners”

Line 345 - 347: The paragraph should discuss the strength of the study, not the strength of RQI.

---

## Round 0.3 · accepted · Accept

Thank you for your patience with the reviewing process. Your submission has been accepted for publication.

Reviewer 1 ·

Basic reporting

no comment

Experimental design

no comment

Validity of the findings

no comment